# Obstacles to Glioblastoma Treatment Two Decades after Temozolomide

**DOI:** 10.3390/cancers14133203

**Published:** 2022-06-30

**Authors:** João Victor Roza Cruz, Carolina Batista, Bernardo de Holanda Afonso, Magna Suzana Alexandre-Moreira, Luiz Gustavo Dubois, Bruno Pontes, Vivaldo Moura Neto, Fabio de Almeida Mendes

**Affiliations:** 1Instituto de Ciências Biomédicas, Universidade Federal do Rio de Janeiro. Av. Carlos Chagas Filho 373, Centro de Ciências da Saúde, Bloco F, Ilha do Fundão, Cidade Universitária, Rio de Janeiro 21941-590, Brazil; joaovictor.roza@ufrj.br (J.V.R.C.); carolinabatista@icb.ufrj.br (C.B.); bernardoholanda@ufrj.br (B.d.H.A.); bpontes@icb.ufrj.br (B.P.); vivaldo@icb.ufrj.br (V.M.N.); 2Instituto Estadual do Cérebro Paulo Niemeyer, Rua do Rezende 156, Rio de Janeiro 20231-092, Brazil; 3Instituto de Ciências Biológicas e da Saúde, Universidade Federal de Alagoas, Campus A.C. Simões, Avenida Lourival Melo Mota, Maceio 57072-970, Brazil; magna.suzana@propep.ufal.br; 4UFRJ Campus Duque de Caxias Professor Geraldo Cidade, Rodovia Washington Luiz, n. 19.593, km 104.5, Santa Cruz da Serra, Duque de Caxias 25240-005, Brazil; gustavodubois@caxias.ufrj.br

**Keywords:** glioblastoma, brain tumor, cancer stem cells, molecular oncology, chemoresistance

## Abstract

**Simple Summary:**

Glioblastomas are the most common and aggressive brain tumors in adults, with a median survival of 15 months. Treatment is surgical removal, followed by chemotherapy and/or radiotherapy. Current chemotherapeutics do not kill all the tumor cells and some cells survive, leading to the appearance of a new tumor resistant to the treatment. These treatment-resistant cells are called tumor stem cells. In addition, glioblastoma cells have a high capacity for migration, forming new tumors in areas distant from the original tumor. Studies are now focused on understanding the molecular mechanisms of chemoresistance and controlling drug entry into the brain to improve drug performance. Another promising therapeutic approach is the use of viruses that specifically destroy glioblastoma cells, preserving the neural tissue around the tumor. In this review, we summarize the main biological features of glioblastoma and the therapeutic targets that are currently under study for new clinical trials.

**Abstract:**

Glioblastomas are considered the most common and aggressive primary brain tumor in adults, with an average of 15 months’ survival rate. The treatment is surgery resection, followed by chemotherapy with temozolomide, and/or radiotherapy. Glioblastoma must have wild-type IDH gene and some characteristics, such as TERT promoter mutation, EGFR gene amplification, microvascular proliferation, among others. Glioblastomas have great heterogeneity at cellular and molecular levels, presenting distinct phenotypes and diversified molecular signatures in each tumor mass, making it difficult to define a specific therapeutic target. It is believed that the main responsibility for the emerge of these distinct patterns lies in subcellular populations of tumor stem cells, capable of tumor initiation and asymmetric division. Studies are now focused on understanding molecular mechanisms of chemoresistance, the tumor microenvironment, due to hypoxic and necrotic areas, cytoskeleton and extracellular matrix remodeling, and in controlling blood brain barrier permeabilization to improve drug delivery. Another promising therapeutic approach is the use of oncolytic viruses that are able to destroy specifically glioblastoma cells, preserving the neural tissue around the tumor. In this review, we summarize the main biological characteristics of glioblastoma and the cutting-edge therapeutic targets that are currently under study for promising new clinical trials.

## 1. Introduction

### 1.1. Overview of Glioblastoma Classification, Mutations and Treatment 

More than 120 types of central nervous system tumors have been described. In spite of the high variability, these tumors do not always represent a diagnosis of malignancy as only 32% of them are malignant. Among brain tumors, there are those originating from glial cells, known as gliomas. Glioblastoma (GBM), a high-grade subtype of glioma, presents astrocytic differentiation and stands out as the most common and aggressive primary brain tumor in adults, representing more than 50% of all gliomas. According to the World Health Organization (WHO), there is no prevention protocol for GBMs, and most patients only become aware of the tumor with the onset of the first symptoms, such as cognitive impairments, neurological deficits and headaches [1]. These symptoms often emerge when the tumor is already properly installed and cell proliferation has escaped the control steps. Despite treatment, only 5% of patients survive for more than 5 years [2,3].

Historically, histopathological analysis considers GBMs as astrocytomas due to their high morphological similarity to astrocytes. In hematoxylin–eosin (H&E) staining, it is possible to detect polygonal or spindle-shaped cells with irregular borders and an acidophilic fibrillar cytoplasm, nuclear atypia, microvascular proliferation with newly developed vessels, necrotic areas and high mitotic activity. These characteristics constitute their main histopathological features in which pathologists rely for diagnosis [4,5,6,7].

However, new classifications have emerged, especially after technical advances in molecular profiling. The 2021 WHO classification takes histological and molecular characteristics in consideration and divides the diffuse adult gliomas into three subtypes: (1) GBM IDH-wild type, (2) astrocytoma IDH-mutant and (3) oligodendroglioma IDH-mutant 1p19q co-deleted. This new classification differs from the previous 2007 WHO classification because it takes into account analysis of mutations in Isocitrate dehydrogenase 1 (IDH) gene, in Telomerase reverse transcriptase (TERT) promoter and co-deletion of chromosomal regions 1p and 19q [8]. Thus, the 2021 WHO classification states that GBMs must have wild-type IDH gene and at least one of the following characteristics: microvascular proliferation, necrosis, TERT promoter mutation, EGFR gene amplification or combined copy number alteration of chromosomes 7 and 10 [9]. 

The discovery of IDH mutations configurated a major breakthrough in the glioma field. The genetic status of IDH is currently one of the most important aspects that have to be taken into consideration to evaluate the prognosis and treatment of different subclasses of gliomas [9,10]. In the 2021 WHO classification, GBM is no longer divided into IDH-mutant and IDH-wild type. IDH-mutant tumors are now classified as astrocytomas or oligodendrogliomas depending on other markers and, in general, the presence of IDH-mutant genes confers a better prognosis and an increase in life expectancy for the patient. The importance of IDH and the consequences of having this mutation will be further discussed in a separate topic because it is still an important line of investigation for the discovery of treatment strategies against GBM. 

Another frequent mutation in GBM is the one that occurs in the TERT promoter. This gene encodes the subcatalytic unit of telomerase, an enzyme involved in the “reconstruction” of telomeric subunits [11,12]. This is the most common mutation in GBMs and it results in increased expression of telomerase, present in more than 69% of all GBMs. The physiological meaning of this mutation is extremely important due to its impact on cell proliferation. Since telomere attrition is one of the key signals to senescence, the ability to stably elongate these structures ensures cell immortality and tumor maintenance [12,13,14].

The most common chromosomal abnormalities found in GBMs are gain of chromosome 7p (trisomy) and loss of chromosome 10q (monosomy). One of the oncogenes on 7p is *Epidermal Growth Factor Receptor (EGFR)*, which is amplified in about one-third of GBMs, and in 10q there is the inactivation of tumor-suppressor genes in the telomeric region of 10q, especially *Phosphatase and Tensin Homolog (PTEN**)* [15]. The large number of brain tumor subtypes and the multiplicity of mutations do not always reflect in a variety of treatments, which are multimodal, consisting of maximum surgical resection, radiotherapy and chemotherapy. For GBMs, this same treatment is maintained in most patients, using the chemotherapeutic drug temozolomide (TMZ) [16].

TMZ is an alkylating agent, applied in the clinic since 2005 as the gold-standard treatment for most brain tumors, including GBMs. Once inside the cell, the TMZ molecule undergoes conversion to monomethyl-triacene-imidazol-carboxamide (MTIC), which acts on DNA and is based on the alkylation capacity of the O6 and N7 positions of guanine, methylation and, consequently, induction of cell cycle arrest during the G2/M phase, before completing cell replication [17,18].

GBM has some characteristics that increase its malignancy leading to a poor prognosis, such as chromosomal aberrations, chemoresistance, high proliferation and intrinsic and extrinsic heterogeneity, caused mainly by subpopulations of tumor stem cells, in addition to being highly infiltrative and modulating the environment around it, recruiting immune cells and modifying the extracellular matrix (Figure 1).

Several compounds are being tested to treat GBM, mostly natural compounds, which have been successful in in vitro approaches. Conjugated treatments with TMZ and new compounds have also been explored, with promising results, since TMZ was approved for clinical use in 2005 and there is no other chemotherapeutic agent for GBM therapy [19].

The gold-standard treatment for GBM only provides an average of 15 months’ survival for patients. Several other therapies, including those acting on GBM cytoskeletal proteins, GBM stem cells and IDH mutations, together with other strategies such as those to overcome the blood–brain barrier or to use oncolytic virus have been widely tested. The next sections of this review will highlight each of these strategies, presenting the related current knowledge and future perspectives. 

### 1.2. GBM Stem Cell Population: Implications in Tumor Resistance and Recurrence

GBMs are tumors that show great heterogeneity at cellular and molecular levels. They present a very complex and diversified molecular signature among different patients and also an intrinsic cellular heterogeneity of each tumor mass, formed by cell niches with distinct phenotypic characteristics. It is believed that the main responsibity for the emergence of these niches belongs to subpopulations of tumor stem cells (TSC), also called tumor initiating cells [20]. They have been classified as a tumor cell subpopulation capable of tumor initiation and asymmetric division. Like normal stem cells, TSCs have the capacity of self-renewal, to originate several phenotypically different cell types and to establish contact with the microenvironment, in addition to having specific stem cell markers. GBM initiating cells are also capable of forming oncospheres from a single CD133+ cell and generate a tumor when at least 100 cells are transplanted into the brain of immunodeficient mice, which does not occur with CD133- cells [21,22,23]. It has been shown that these cells are resistant to several chemotherapeutic drugs such as TMZ, carboplatin, paclitaxel (Taxol) and etoposide (VP16), as well as radiotherapy [24,25]. The wide variety found among GBMs from different patients in relation to the percentage of tumor stem cells is remarkable. Some articles found that samples of the primary GBM tumor mass obtained from three different patients had between 10 and 69.7% of CD133+ cells. These cells express 32 to 56 times more of the MGMT protein responsible for DNA repair and escape from TMZ treatment [25]. Furthermore, anti-apoptotic genes, including *FLIP*, *BCL-2* and *BCL-XL*, were also found at higher levels in CD133+ cells than in CD133- cells from the same tumor mass [25]. 

Due to the high cellular heterogeneity of GBM, 40% of primary GBM samples do not have CD133+ cells, however, they have cells within the tumor mass with characteristics of stem cells. Thus, Son and colleagues screened neural stem cell markers in 24 GBM samples and identified CD15 as a marker for another GBM stem cell lineage that is CD133- [26]. 

The microenvironment of GBM stem cell populations plays a crucial role in tumor maintenance. A subpopulation of GBM stem cells, preferentially located in perivascular regions, expresses high levels of CD44 and ID1 [27]. Another highly expressed marker found in such subpopulations close to perivascular regions is the α6 integrin. This marker may co-localize with CD133, however, even when cells are negative for CD133, those cells still have a high capacity to generate tumors. Knockdown of α6 integrin with shRNA inhibits cell growth, induces apoptosis in culture and inhibits tumor formation in vivo [28].

Assuming that neural stem cells lose their self-renewal capacity when differentiated into neuronal-like or glial-like phenotypes, one of the strategies to limit tumor growth is to induce tumor stem cells to a postmitotic mature cell fate. Bone morphogenetic protein (BMP) is a member of the TGF-β superfamily capable of inducing differentiation of GBM stem cells into astrocytes [29,30]; however, astrocytes are able to re-enter in the cell cycle, again inducing tumorigenesis. Achaete-scute homolog 1 (ASCL1) is a growth factor that participates in the neural differentiation process during embryonic development. A subpopulation of GBM tumor mass cells expressing high levels of ASCL1 has been found [31]. Patients with high levels of ASCL1 had longer survival than patients with low expression of ASCL1. ASCL1 is able to induce GBM stem cell differentiation, decrease proliferation rate, increase neurospheres formation in vitro, and tumor mass growth in vivo is inhibited when it is overexpressed [31].

Given the high rates of tumor recurrence and low patient survival with conventional treatment, new therapeutic approaches targeting TSCs are being proposed and developed. The aim of new therapeutic approaches is to combine classic with new drugs that target TSCs [32]. Thus, its recurrence with a more malignant phenotype in patients is avoided, increasing their survival. 

## 2. IDH Mutation and Resistance Outcome

Parsons and colleagues were the first to identify IDH genetic alteration in GBM specifically [33]. A study of genome-wide exon sequencing was performed in 22 GBM samples from different patients, and one of the biggest surprises was the discovery of this novel mutation, which resulted in the replacement of an arginine 132 by one histidine in amino acid 132 of IDH1 (R132H IDH1 mutation). R132H IDH1 was present in 12% of the GBM samples, suggesting an important role in glioma transformation process [33].

The IDH1 gene is located at chromosome 2q33 and encodes IDH1 protein which resides in the cytoplasm [34]. IDH1 catalyzes the oxidative carboxylation of isocitrate to α-ketoglutarate (α-KG), which is noteworthy involved in the activity of tricarboxylic acid cycle (also known as Krebs cycle or citric acid cycle) [35]. Further works showed that the R132H IDH1 mutation was also a frequent feature of acute myeloid leukemia (AML), encountered in 16% of the studied samples [36]. 

A mutant IDH2 gene (IDH2 R172K or R172M) has additionally been identified in glioma subsets, although less frequently than IDH1 mutations [37]. IDH2 is analogous to IDH1. Like IDH1, IDH2 catalyzes the formation of α-KG from isocitrate but is localized in the mitochondria. A spectrum of mutations is observed at IDH1 and IDH2 in cancers. For instance, common IDH1 mutations include R132H and R132C, and common IDH2 mutations include R172K and R172M [38].

Almost all reported cases of gliomas bearing IDH mutations have been heterozygous, and inactivating alterations such as deletions or nonsense mutations were not observed for this gene in any cancer. The evidence of a possible genetic selection led to the conclusion that the IDH mutations confer the enzyme with an oncogenic gain of function [39].

Zhao and colleagues reported a dominant-negative role for mutant IDH1 R132H, the function of the wild-type enzyme being inhibited in vitro in presence of the mutant enzyme [39]. They additionally showed that IDH1/2 wild-type isoforms prevented the accumulation and overexpression of the hypoxia-inducible factor-1α (HIF-1α), a transcription factor associated with aggressive types of cancer [39]. These results led the authors to suggest a tumor suppression function for IDH1/2. In parallel studies, Yan and colleagues showed that IDH1/2 mutations reduced the enzyme’s ability to generate α-KG [37]. Dang and colleagues provided the missing piece of the puzzle, showing that IDH1/2 mutants gain the neomorphic enzymatic activity to reduce α-KG to R(–)-2-hydroxyglutarate (2HG) [40]. Consistent with this finding, measurement of 2HG in IDH1/2 mutated gliomas or leukemia samples revealed 2HG levels higher than in neoplasms bearing wild-type IDH1/2 [40,41]. The fact that tumors with IDH1/2 mutations overexpressed HIF-1α when compared to wild-type IDH1/2 tumors suggested that 2HG could act as an oncometabolite, promoting malignant transformation.

The IDH1/2 mutations are a dividing point to categorize gliomas. Virtually all tumors presenting wild-type copies are classified as GBMs. On the other hand, gliomas harboring IDH mutations can be subdivided into two subclasses: those presenting chromosome 1p/19q co-deletion, referred historically as oligodendrogliomas; and those without 1p/19q co-deletion, representing astrocytomas [42]. The presence or absence of the co-deletion determines the clinical evolution whereas around 94% of IDH-mutant non-1p/19q co-deleted gliomas present TP53 mutations and 86% present inactivating ATRX mutations [42]. These genetic alterations, highly frequent in such tumors, strongly suggest that IDH mutation is likely to be at the origin of such malignancies and could give rise to tumors of both lineages astrocytic and oligodendrocytic. Clinically, IDH mutation is a favorable prognostic factor when compared to wild-type gliomas, as patients respond better to chemotherapy as explained next [37].

The wild-type IDH1/2 catalyzes the conversion of isocitrate to α-KG and, at the same time, the reduction of NADP+ to NADPH with the production of CO_2_. They can promote the formation of hydrogen bonds with the β-carboxyl of isocitrate. Mutations replacing the R132 of IDH1, as well as the R140 and R172 of IDH2, confer to the enzymes the ability to convert α-KG into 2HG concomitantly with the oxidation of NADPH to NAD+. 2HG and α-KG share a very similar chemical structure, but differ only in the presence of a hydroxyl group in 2HG at the same carbon that establishes a bound with an oxygen atom in the α-KG molecule [40].

The discovery of these intriguing data involving the neomorphic activity of IDH1/2 raised the question of the mechanisms by which enhanced 2HG levels could lead to the accumulation of HIF-1α. Several studies from different laboratories have shown that 2HG impairs the catalytic function of α-KG-dependent enzymes. It acts as a competitor of α-KG, occupying its binding site in these enzymes, a competition authorized by the structural similarity of 2HG to α-KG [43,44]. Interestingly, prolyl hydroxylases (PDH) are among the enzymes susceptible to have their activity modified by 2HG. Those enzymes are responsible for targeting HIF-1α to degradation [45], an activity consistent with the high levels of HIF-1α observed in IDH mutated tumors [39]. PHD inhibition by 2HG remains however debated. Koivunen and colleagues have indeed provided convincing data showing decreased HIF-1α levels in cells treated with 2HG [46]. PHDs are not the only enzymes that can potentially be affected by the overproduction of 2HG. 2HG accumulation due to IDH mutated isoforms has additionally been shown to inhibit the catalytic activity of the ten-eleven translocation (TET) enzyme family. TET family components are key mediators of DNA demethylation and upon its inhibition, 2HG accumulation produces a hypermethylated profile in such cells. [47,48,49]. This finding raises 2HG to a whole new level, because it implies a role for this metabolite in gene expression control, and brings forward a role for 2HG and α-KG-dependent enzymes in epigenetic control of cancer cell behavior.

Similar to 2HG, other metabolites have been shown as potent modulators of epigenetic-related enzymes, especially those that depend on α-KG. Recently, the metabolite gamma-hydroxybutyrate (GHB) was demonstrated as an inhibitor of TET2 enzymes in GBM stem-like cells (GSC) [50]. The authors have shown that more differentiated glioma cells decrease the expression of succinic semi-aldehyde dehydrogenase (SSADH), an enzyme involved in GABA metabolism [51]. Upon this downregulation, GHB is overproduced and acts as a competitor to α-KG resulting in a hypermethylation profile and a decrease in proliferation and tumorigenic properties of GSC [50].

Elucidating the function of 2HG in cancer cells is further complicated by the occurrence of enantiomers. High levels of the right-handed enantiomer of 2HG, D-2HG, have been reported to cause oxidative stress in rat brains [52], an event that could potentially promote oncogenesis. In the human clinics, high levels of 2HG have been linked to a rare metabolic disorder known as D-2-hydroxyglutaric aciduria [53,54], resulting in most cases from a mutation in D-2-hydroxyglutarate dehydrogenase. Although these patients showed significantly higher levels of D-2HG compared to leukemia or glioma patients harboring IDH1/2 mutations, they do not have a predisposition to developing gliomas, leukemia or other malignancies [55]. Moreover, patients bearing gliomas with wild-type isoforms of IDH1 present a poorer prognosis when compared to patients with IDH1 mutated tumors [33]. Even with the demonstration that an accumulation of D-2HG, in the case of the D-2-hydroxyglutaric aciduria, does not predispose individuals to develop tumors and those patients’ bearing gliomas with IDH1 mutations have better survival expectancies than those bearing the wild-type ones, the overproduction of 2HG in gliomas, driven by the IDH mutations, has been used as an argument to explain the genesis of glioma.

As a byproduct derived from the TCA cycle, 2HG is among the metabolites that have been shown to be important in the progression of gliomas. Notably, they change glioma physiology by interfering in the functions of αKG-dependent enzymes, impairing epigenetic modifications. The discovery of the influence of 2HG and other metabolites in tumor behavior, together with data acquired in genetics, brought back the cancer metabolism as one of the major effectors in tumorigenesis, rendering important the study of the reactions involved in the cellular reprogramming of energy metabolism pathways.

The discovery of mutations in IDH1 and 2 revolutionized the treatment of malignancies bearing such genetic alterations. They have been observed in multiple types of cancer such as gliomas, AML, cholangiosarcomas, myelodysplastic syndromes and chondrosarcoma. The FDA-approved inhibitory drugs used in trials in the last few years are Ivosidenib [56] for IDH1 mutations and Enasidenib in the case of IDH2 mutations [57]. Despite the relative success of these small drugs in the treatment of AML, they have failed to treat gliomas and other types of IDH mutated cancers. New drugs are in development, such as Vorasidenib, a dual inhibitor of mutant IDH1 and 2. Results of a phase I clinical trial have shown a favorable safety profile for treating gliomas and tumor shrinkage in patients with non-enhancing gliomas [58]. These data are promising and perhaps IDH inhibitors can become a pathway to treat such disease. Beyond small molecule inhibitors, recently, it was reported the development of a new vaccine against gliomas carrying IDH1 mutation. Platten and colleagues have demonstrated that a vaccine against the IDH R132H isoform was able to induce immune response in 93% of IDH mutant glioma cases. Moreover, they reported high three-year progression-free and death-free rates [59]. 

Accumulation of evidence regarding the metabolic and physiologic consequences of IDH mutations has enabled scientists to advance in new therapeutic avenues, but it is still necessary to understand how they are acquired and what are the selective advantages that such mutation brings to cells. 

## 3. The Cytoskeleton as a Potential Therapeutic Target against Glioblastoma

Another broad group of therapeutic targets against GBM is cytoskeletal proteins. The cytoskeleton is composed by three main components: microfilaments, intermediate filaments, and microtubules, together with their accessory proteins. Its major functions are to sustain the shape of cells and also to allow cells to resist against mechanical deformations. The cytoskeletal proteins span throughout the cytoplasm, cell nucleus and plasma membrane, connecting and integrating intracellular structures with the extracellular environment. As it is involved in several cellular processes, such as transport of molecules, cell migration, differentiation and proliferation, abnormalities in its structure and function are often the cause of many diseases [60].

The highly infiltrative and recurrence capacities of GBM are strongly associated with aberrant expression of several cytoskeletal proteins [61,62]. Thus, agents that act in the cytoskeleton have been used against GBM due to their capacity to interfere with cell proliferation [63]. In this part of the review, we will focus on the role of cytoskeletal proteins and structures as targets in GBM therapy. 

### 3.1. Microtubules as Therapeutic Targets against Glioblastoma

Microtubules are essential cytoskeletal elements in mitosis and cell division, therefore, are one of the best and most important targets in several cancer therapies. Drugs that target microtubules are mainly divided into microtubule destabilizing and stabilizing drugs [64].

Among microtubule stabilizing drugs, Paclitaxel (also known as Taxol) and its different derivatives has been tested for GBM therapy. The soluble form of the drug was extensively tested against GBM, but it turned out to be unable to properly penetrate the blood–brain barrier [65]. Thus, improvements have been developed such as other forms associated with copolymers made of PLGA (poly (d,l lactide-co-glycolide)) and PEG (polyethylene glycol) [66], conjugated with other molecules, like poly-L-glutamic acid [67,68] or the peptide Angiopep-2 [69]; however, the clinical trials using these Taxol variations have been terminated or have yet to yield conclusive results. 

Other taxane analogue molecules such as Ortataxelis [70], Cabazitaxel [71] and TPI-287 [72] have been tested. While the first two drugs did not show significant effects on patients’ survival, the third drug was well-tolerated and three patients out of seven had partial response. Epotilones is another class of tubulin stabilizing drugs that have also been used against GBM. Patupiloneis [73], Sagopilone [74] and Ixabepilone [75] have been tested in phase I/II clinical trials but there were mild or no significant effects on patients’ survival. 

Like microtubule stabilizing drugs, several microtubule destabilizing drugs are being explored in GBM treatment. Vincristine has been used in cancer treatment together with procarbazine and lomustine forming what is called the PCV treatment. However, in a large clinical trial, 447 patients with recurrent glioma were treated with PCV but no significant effects in patients’ survival were observed [76]. Other drugs, such as Verubulin [77], Batabulin [78] and 2-methoxyestradiol [79], also had little or no efficacy in patients’ survival. On the other hand, Lexibulin (CYT997) [80], Lisavanbulin (BAL101553) [81] and Mebendazole [82] have shown promising results in vitro and in mouse models but no results from clinical trials have been released yet. 

### 3.2. Intermediate Filaments as Therapeutic Targets against Glioblastoma

Apart from microtubules, another therapeutic target against GBM could be the intermediate filaments. Indeed, effects on vimentin have been studied. Withaferin-A, a steroidal lactone compound isolated from *Withania somnifera*, is an inhibitor of a variety of proteins, but its most studied target is vimentin [63]. In vitro studies showed that Withaferin-A interferes with the migratory capacity of U251 and U87 GBM cell lines [83]; however, other basic studies are still needed before starting clinical trials.

Pritumumab, an anti-vimentin antibody, shows specificity for GBM but does not show specificity for normal adult cells (neurons, astrocytes or other cells) [84]. This is due to the fact that Pritumumab binds to cell surface vimentin, only expressed in glioma cells. This antibody has been tested in clinical trials to treat patients diagnosed with GBM, astrocytoma or neuroblastoma [84]. In general, Pritumumab showed a 5% increase in patients’ survival. Thus, administration of higher doses of Pritumumab has the potential to improve the life expectancy in GBM patients, although side-effects may appear and are to be determined using other clinical trials. 

### 3.3. Other Alternative Cytoskeletal Therapeutic Targets against Glioblastoma

A new mode of GBM therapy is the application of low-intensity (1–3 V/cm) and alternating electric field of the order of 100–300 kHz applied via cutaneous arrays to provide optimal tumor site coverage [85].

Although the tumor treating fields (TTF) therapy was demonstrated to inhibit cancer proliferation by interfering with the mitotic spindle, it is now clear that TTF act in several other biological processes, including DNA repair, permeability of cell membranes, and intracellular molecules known as dipoles. Specifically, for GBM, TTF is delivered at an optimal frequency of 200 kHz and intensity of 1–2 V/cm [86]. Indeed, a clinical trial using this therapy against GBM was the first, after the introduction of temozolomide chemotherapy [19], to show an increase in patients’ survival [87,88] without any strong systemic adverse events [89]. 

The main cellular and molecular mechanism through which TTF therapy acts is during mitosis, particularly in the mitotic spindle. When TTFs are applied, tubulin proteins tend to align with the electric field, thus interfering with the microtubule polymerization/depolymerization, resulting in spindle malformation and cell cycle arrest. Moreover, failure of the spindle assembly checkpoint [90] leads to incorrect chromosome segregation and possibly cell death [90]. TTF has also been described as capable of interfering with the septin protein complex, known to participate in the cleavage furrow [91,92], again leading to anomalous chromosomal segregation and possibly structural changes to the membrane of cells [92].

The use of TTF therapy still has some disadvantages such as the difficulty to treat more diffuse tumors located at different sites into the brain and also the high costs of treatment per month per patient [93]. Despite these issues, TTF has become “the fourth cancer treatment modality” against GBM [94].

A new mechanism of intercellular communication known as tunneling nanotubes (TNTs), has been described to play a pivotal role in cell–cell interaction [95,96]. TNTs are thin, open-ended, actin-based membrane nanotubes allowing the exchange of various molecules or even organelles, as bridges between cells [97,98].

TNTs were first identified by Rustom et al. in pheochromocytoma PC12 cells [95]. Since then, a variety of in vitro studies have reported the presence of TNTs in several cell types, including GBM [99,100,101,102]. Moreover, reports have shown that communication between GBM and other central nervous system cells such as astrocytes [102,103] and microglia [100] can occur through TNTs. The transfer of mitochondria appears to modulate GBM drug-resistant state [104]. Indeed, it was recently described the capacity of TNTs to transfer mitochondria between GBM stem cells in organoids [105]. It was also demonstrated that TNTs allow GBM cells to adapt to temozolomide and ionizing radiation treatments through transmission of the MGMT protein from the MGMT mRNA-positive cells to other cells expressing low MGMT levels [106]. 

Thus, TNTs seem to be very important for GBM progression and treatment resistance. In the next years, a better comprehension of TNT functions may allow the development of new therapies against glioblastoma. 

## 4. Strategies to Overcome the Blood–Brain Barrier in GBM Treatment

There are three main challenges that need to be overcome for GBM treatment [107]: the cellular heterogeneity within the tumor, which facilitates the selection of resistant subpopulations, the fortified location of tumor which hinders delivery of therapeutics, as well as the induction of a strong local immunosuppression that promotes immune evasion and limits the efficacy of emerging immunotherapies [108] (Figure 2).

The blood–brain barrier (BBB) is a complex neurovascular structure responsible for the control of neural homeostasis through selection of molecules that can penetrate the nervous tissue [109]. It is composed of brain endothelial cells, pericytes, astrocytes, microglial and smooth muscle cells. The BBB is a unique structure where tight junctions and absence of fenestrations result in an almost impenetrable barrier essential to protect the brain from toxins and other harmful substances. However, it also works as a hermetic barrier preventing the success of therapies directed to brain disorders by selection of drugs [110].

Molecules that can cross the BBB by simple diffusion must have lipophilic characteristics and/or weight less than 500 Da [109,111]. Several studies and experimental therapies seek to improve drug delivery into the tumor mass through a transient opening of the barrier, but most have failed since the integrity of the BBB cannot be disrupted [112,113,114]. Experiments show that transient opening of the BBB increases the input of chemotherapeutic agents in the tumor mass. In addition, it allows the use of chemotherapeutics that do not pass or barely surpass the barrier, such as doxorubicin (DOX) and topotecan, as adjuvants to combat GBM alone or in combination with TMZ [112,115].

Current approaches, under development, aim to circumvent the BBB through its transient opening using chemicals, osmotic agents, “molecular trojan horse-related methods”, physical methods, such as “focus-ultrasound” (FUS), radiotherapy or BBB bypass strategies, such as “convection-enhanced delivery” (CED), drug-impregnated microchips, “intra-arterial brain infusion”. All approaches involve high costs, specialized equipment or need the conjugation of drugs to vectors for specific targeting [109,112,116,117].

In the search for a transient opening of the BBB, it has been tested its biochemical modulation with drugs such as agonists of the kinin pathway, peroxynitrites, NF-kB, VEGF, EGFR, among others, that increases BBB permeability via paracellular transport [111,112,114,116,118]. Bevacizumab is a monoclonal antibody designed against VEGF that has been shown to reduce tumor growth, brain edema and promote GBM tumor regression, but its combination with TMZ and radiotherapy has failed to increase overall survival [119]. In general, bevacizumab is very effective in reducing edema and swelling and alleviate symptoms for a short period of time with some improvement in quality of life, but it does not induce an antitumor effect or prolong survival [119]. Lu and colleagues [120] provided one piece of evidence that may help elucidate why bevacizumab is not effective in prolonging patients’ survival. They have demonstrated that VEGF signaling pathway directly inhibits epithelial–mesenchymal transition in GBM cells through the recruitment of the heterocomplex MET/VEGFR2. The use of the monoclonal antibody anti-VEGF blocks this inhibition, increasing tumor cells’ ability to migrate and create micrometastasis in brain areas that were previously healthy [120]. 

Another target that promotes angiogenesis and is up-regulated in GBM is protein kinase C (PKC). In preclinical and clinical trials, the PKC inhibitors enzastaurin and tamoxifen have shown promising results, increasing cell death [121]. On the other hand, cetuximab, a monoclonal antibody that inhibits EGFR signaling, and other inhibitors of this receptor presented problems in the evolution from basic research to clinical trials, among them caused by the size of the molecule and low permeability by the BBB [122].

High-intensity FUS is a methodology that is being used in clinical trials to non-invasively permeabilize the BBB for systemic therapeutic delivery to GBM; it uses ultrasound waves in conjunction with systemic administration of microbubbles, a contrasting agent, and holds promise in preclinical and clinical trials [116,123,124]. Oscillations of microbubbles in blood vessels create shear stress and rapid collapse of microbubbles, reducing tight junction integrity, in addition to induction of local inflammatory process [116]. The transient opening of the BBB promotes an intratumor increase in TMZ, doxorubicin (DOX), cisplatin, paclitaxel and docetaxel (DTX) concentrations, alone, or in combination with nanoparticles [124,125]. Advances in technical protocols, which still require a bone window in the skull, are constantly evolving to make the FUS more practical for the treatment of gliomas [109].

Radiotherapy, widely used after surgical resection to eliminate residual tumor cells, has been used as a tool for barrier permeabilization and transport of pharmacological molecules, such as sub-nanometer-sized gold particle compounds with effective antitumor drugs [126]. Skipa [112] showed that radiotherapy promotes the opening of the BBB, which seem to be important for the permeabilization of the barrier, being maintained long after radiation. Radiotherapy of brain cancer induces acute and late pro-inflammatory reactions that involve the production of several cytokines (e.g., TNF-α, IL-1, IL-6, IL-8, IFN-γ), many of which are potent inducers of kinin expression [127].

CED is a local-targeted drug delivery to the CNS, with a microcatheter placed directly into the GBM tumor through minimally invasive surgery [109]. The main goal is to bypass the BBB, allowing the delivery of high-dose therapeutics for large volumes of interest and limiting associated systemic toxicities. CED has the advantage of not limiting the molecular weight or lipophilicity of the substance, which allows the testing of numerous agents, including chemotherapy drugs, toxins, monoclonal antibodies, genes, viruses and nanomedicine. However, the surgical procedures (such as backflow of the injected fluid) and the mechanisms of action of the tested agents, combined with the additional costs of conducting a trial evaluating CED, limited the studies to results without significant data in phase II or III [109,123,128].

A model not yet tested in humans for the treatment of GBM, drug-impregnated microchips, represents a strategy to circumvent the BBB and control the pharmacokinetic properties of one or more drugs with high precision using an external stimulus. Microchips work as drug reservoirs since the release is controlled by an external stimulus such as an electrical pulse. These microchips have not yet been tested in humans, but the in vivo trials using rat models showed encouraging results [129].

A wide range of spherical and non-spherical nanoparticles (NPs) with different sizes, architectures and surface properties have also been designed as drug delivery systems for the brain. Among them, lipid-based nanoparticles, such as liposomes, have been identified as promising in the delivery of antitumor drugs to the CNS, overcoming the limitations imposed by the BBB, but they need improvement to more efficiently entrap the drug and be stable in the long term [111,125,130]. Liposomes enriched with monoclonal antibodies against transferrin receptors, GFAP or GLUT4, increased DOX transport to the brain [109].

NPs in pre- and clinical trials usually range from 1 to 100 nm and can incorporate different pharmaceutical agents [131] and aim to combat side effects and increase the demand for drugs in tumor tissue, such as nanoparticles coated with anti-VEGF Ig, molecules being used in preclinical studies [132]. In addition to liposomes, other NPs are being tested and approved for non-brain cancers, such as polymer NPs, solid lipid NPs, magnetic NPs, quantum dots, porous silicon NPs, mesoporous silica NPs among others [115,126,131,133]. Recently, it has been shown that intranasal administration of nanoparticles enriched with chemotherapeutic molecules can be an efficient strategy to bypass the BBB and allow entry of drugs into the brain via the olfactory and trigeminal nerves [130]. The literature indicates clinical trials combining CED and liposomes for intracellular drug delivery of agents normally unable to cross cell membranes [128].

In addition to nanoparticles, cell-penetrating peptides (CPPs) are promising tools for drug delivery for antitumor treatment. These peptides are short (usually less than 30 amino acids), positively charged, with amphipathic characteristics and are able to cross the BBB with high efficiency. However, the lack of specificity in cell type remains one of the main disadvantages for its clinical development [134].

In addition to the development of new technologies that could improve GBM treatment, current strategies are not effective since the only chemotherapeutic agent wildly used in clinic is TMZ, and although TMZ is capable of transpose BBB, its intratumoral concentration is low, and there are some cellular subpopulations that are resistant to TMZ and promote tumor recurrence in more malignant and aggressive phenotypes. 

## 5. Using Oncolytic Viruses to Reach Tumor Cells

Recently, attenuated viruses have been evaluated for the treatment of GBM. In general, viruses used to treat tumors can be divided into three categories: replication-deficient viruses, such as retroviruses or adenoviruses, whose therapeutic potential is derived from the gene(s) they deliver; oncolytic viruses, such as herpes simplex virus (HSV) or adenovirus, which selectively divide inside tumor cells and then lyse and kill the infected cell before releasing progeny viruses; and replicating nonlytic viruses such as retroviruses, which selectively divide inside tumor cells while releasing progeny viruses in a nonlytic fashion, with the primary therapeutic effect occurring from the gene(s) they deliver [135]. 

OVs have gained space in new therapy proposals not only for their ability to induce tumor cells lysis, but also for increasing the response to immunotherapies through their combination with immune checkpoint inhibitors, and, recently, they have been studied as potential vectors for gene therapy in difficult-to-access tumors [108].

The idea is that OVs would only replicate in tumor cells, but not in normal cells, due to specific tropism. Once infected, tumor cells are susceptible to cell death leading to viral replication and propagation, sustaining this antitumor effect [136]. The infection results in recruitment of innate and adaptative immune response to the tumor. These mechanisms may vary depending on the OV used. Some of them, such as Newcastle disease virus (NDV) and herpes simplex virus (HSV), has both ways: direct cytolysis followed by recruitment of an immune response, once tumor cells are lysed, there is a release of tumor-associated antigens which leads to induction of local and systemic antitumor immunity [137]. 

There are several clinical trials involving OVs. Related to malignant gliomas, NDV [138], reovirus [139], parvovirus [140], adenovirus [141], poliovirus [142], vaccinia virus [143] and HSV [144] are being clinically tested. Among those OVs, HSV is the leading candidate for treating gliomas, being the most wildly characterized with six completed clinical reports. Recently, in 2015, the Food and Drug Administration (FDA) approved T-Vec therapy for melanoma [145], increasing the expectation about its use in glioma therapy. The use of OV for treatment of GBM has shown promise in early-stage trials. Excellent examples are FDA Fast Track Designation of two OVs, PVS-RIPO and DNX-2401, attenuated poliovirus and adenovirus, respectively [142,146,147], and conditional and time-limited approval, from the Japan Ministry of Health, for phase 2 studies in Japan, of DELYTACT (teserpaturev/G47∆), that is a genetically engineered oncolytic HSV type 1 [148].

OVs can be administrated in many routes and dosing protocols. The most used routes of administration in clinical trials are intravenous (i.v), intraarterial (IA), inhalation and intratumoral (IT), with the IT being the most studied so far. IT route maximizes virus concentration, at the site of the tumor inducing robust immunological response, leading to favorable pharmacokinetics, and avoiding the BBB [149].

Direct IT inoculation has demonstrated the most success in the FDA-approved T-Vec therapy as well as the Fast Track Designation for both PVSRIPO and DNX-2401, showing no neurovirulence, and a survival average of 24 to 36 months. IT inoculation overcomes several hurdles of systemic delivery such as BBB penetration, accumulating oncolytic virus at the cancer site and protecting the integrity of the oncolytic virus from the circulating immune system [149]. In contrast, due to cost and complexity of neurosurgical procedures, repeated IT inoculation is not often feasible. Deep-seated tumors, or tumors in deep areas of the brain, limit the applicability of IT injection [136].

Intravascular therapy presents several challenges such as systemic toxicity, immune clearance, that dramatically reduces the efficacy of oncolytic viruses with neutralizing antibodies. BBB penetration also remains a longstanding obstacle [150]. Nevertheless, intravascular oncolytic virus therapy continues to be a promising avenue of investigation given the potential that OVs have to spread throughout the tumor, including the periphery and distant tumor cells. To overcome challenges associated with intravascular OV delivery, methods of circumventing the BBB and neutralizing antibodies have been recently explored: mannitol infusions, focused ultrasound and microbubble-mediated drug delivery systems are under investigation [151]. Gesundheit et al. [152] used mixtures of NDV, parvovirus and vaccinia delivered IA and intravenous. Survival from diagnosis ranged from 4 to 14.5 years.

Oncolytic viruses (OV) have been explored to develop therapeutic gene delivery approaches. In GBM, OV-mediated gene therapy involves the delivery of genetic material to directly kill cancer cells [153]. The problems for the methodology to be applied in the clinic are the high cost, the difficulty to transfect all the tumor cells, the improvement of the cellular response and the innate immunity response when in contact with OV [153]. One example is the production of a recombinant measles virus that maintains oncolytic activity while evading immunity against the virus. This virus was developed for malignant tumors outside the CNS, but it can be useful because it keeps the virus circulating for a longer time [144]. Systemic therapy combined with IT therapy may offer a synergistic effect including a local and systemic immune response [154].

Another promising approach is the use of dendritic cell therapies that are thought to offer particular synergy with oncolytic virus but need further clinical investigation. Since the mid-1990s, dendritic cells have been used in clinical trials as cellular mediators for therapeutic vaccination of patients with cancer, proving to be safe and induce antitumor immunity, even in patients with advanced disease [155].

OVs are potentially interesting tools for GBM treatment due to their specificity and effectiveness. Initial results are promising, and new recombinant OVs are being produced. Additionally, techniques are being used along with OV therapy, significantly increasing the therapeutic possibilities, making them less invasive and more accurate. 

## 6. Development of New Drugs for Glioblastoma Treatment

The conventional treatment for GBM is limited to tumor removal surgery, radiotherapy and chemotherapy. Chemotherapy can work by blocking the blood supply to tumor cells or preventing the cell division process [156]. In 2005, the use of temozolomide (TMZ) was approved by the US FDA [157], based on a clinical study of 573 patients diagnosed with GBM. A 2.5-month increase in median survival was observed in patients treated with TMZ and radiotherapy compared to those treated with radiotherapy alone. Therefore, a significant result was achieved with minimal toxicity [158]. 

In 2009, the FDA also approved the use of bevacizumab (BEV) [157]. From a phase 2 study, it was observed that daily use of TMZ in conjunction with BEV every two weeks was positive and had no significant rejection [159]. Furthermore, in a 2009 clinical review, a median survival increase of 9.2 months was observed for patients treated with BEV alone [160]. 

Recently, a network meta-analysis demonstrated that lomustine appears to be the most effective chemotherapy treatment and other combination therapies tested (BEV monotherapy, BEV plus lomustine, bevacizumab and irinotecan, regorafenib, TMZ plus Depatux-M, and fotemustine) had a higher risk of serious side effects for treatment of first recurrence of GBM [161]. 

Additionally, there are several drugs from various pharmaceutical classes (diabetic agents such as metformin (Table 1); checkpoint inhibitors as nivolumab and pembrolizumab; carbonic-anhydrase inhibitors such as acetazolamide, and immunosuppressive agents as everolimus) that have been studied for drug repurposing in GBM, with interesting results in preclinical studies, but modest in evaluation in early clinical trials, such as those reviewed by Lyne and Yamini (2021) [162].

Importantly, the cell therapies as well as the treatment with chimeric antigen receptor (CAR) T cell represent a new treatment modality with exciting potential, although it has not yet been clinically effective [163]. 

Mifepristone (RU-486) was the first antiprogesterone (progesterone receptor partial agonist) developed and its initial use was as an abortifacient in the first months of pregnancy [164]. In addition, there are several clinical studies using RU-486 (Table 1) for the treatment of cancers [165]. RU-486 binds to both progesterone and glucocorticoid receptors, acting as an antagonist, resulting in a reduction of progesterone-dependent genes (Table 1) [166]. A 1997 study observed an increased expression of progesterone in patients suffering from higher-grade gliomas [167]. Furthermore, a 2001 survey showed that 100% of grade IV and 83% of grade III astrocytomas overexpress progesterone receptors [168]. These analyzes suggest that the expression of these receptors is related to the stage of the glioma, that is, more aggressive tumors have high expressions of progesterone receptors. Currently, several pieces of evidence suggest that RU-486 favors tumor treatment by reducing VEGF and MGMT levels [169]. Furthermore, due to its anti-progesterone and anti-glucocorticoid capacity, it inhibits cell growth, migration and invasion in gliomas [170]. Thus, the crucial role of this drug for the treatment of GBM is evident.

Alternative medicine and natural products are becoming more popular in the treatment of ailments. Flavonoids, which are natural polyphenolic substances found in many of ordinary diets, have been shown to be effective in the treatment and management of various conditions [171]. Due to its varied biological effects, agathisflavone, a plant-derived biflavonoid, has been found to inhibit the growth of a variety of cancer cells [172]. Agathisflavone has also an anti-glioma effect (Table 1). It was demonstrated that agathisflavone decreased cell survival and proliferation in GBM cells in a dosage and time-dependent way while improving cytotoxicity. Moreover, agathisflavone decreased microglia neuroinflammation and reduced microglia and neuron neurotoxicity, demonstrating that this biflavonoid may influence the tumor microenvironment with both mesenchymal tumor and microglia stem cells [173]. Despite looking very promising for use as an anti-glioma drug, agathisflavone has not yet been tested in humans and further studies are still needed to determine its true effectiveness.

Furthermore, 4-Hydroxytamoxifen (OHT) is a well-known estrogen-related receptor antagonist and an active metabolite of the tamoxifen (TMX) prodrug. Previous research suggests that OHT might be used to promote tumor cell death without causing apoptosis (Table 1). Recent studies have investigated the impact of OHT on malignant peripheral nerve sheath tumors (MPNSTs) and discovered an ER-independent inhibition of cell growth. The authors demonstrated and characterized OHT-induced MPNST cell death as nonapoptotic and mediated at least in part by autophagic degradation of KRAS. A study by Graham and colleagues predicted that OHT may elicit comparable death pathways in GBM since MPNST and GBM malignancies are both made mostly of glial cells [174].

The autophagic mechanism has been implicated as a possible chemotherapeutic target for apoptosis-resistant cancers. Given OHT’s pleiotropic effects, it is likely that one or more ER-independent pathways are involved in its cytotoxic impact on hormonally insensitive neoplasms. The mechanism of action of TMX in GBM has long been thought to be Ca_2_ signaling suppression via protein kinase C (PKC). The authors suggested that OHT-induced mortality was highly resistant in a GBM cell line expressing EGFR variant III (EGFRvIII). EGFRvIII was refractory to OHT-induced degradation. As a result, OHT causes GBM cell death via an autophagy-related mechanism, and it should be considered as a possible therapeutic treatment in patients with GBM whose tumors express wild-type EGFR [174]. 

In a clinical trial, the combination of tamoxifen with TMZ was well tolerated, as well as being possibly useful in improving the efficacy of a dose-dense TMZ treatment as a second-line therapeutic approach. Of 32 patients with GBM recurrence (18 men, 14 women; mean age 57 years), the mean time to tumor progression after recurrence (TTP-2) and overall survival time (OS) were 17.5 and 7 months, respectively. There were no variations in OS or TTP-2 between subjects with methylated or unmethylated MGMT. There were no side effects associated with the use of TMZ and tamoxifen in any of the patients. [175].

Despite the use of metformin for diabetes, Yang and coworkers showed that this drug can also be used for glioma treatment (Table 1). According to their research, metformin could reduce TMZ resistance in GBM cells, down-regulating SOX2 expression in vitro, reducing the formation ability of GBM cells, and blocking GBM xenograft growth in vivo [176]. Another important discovery is that the drug can have anti-invasive and antimetastatic potentials. Although the mechanism of action was not cleared, metformin could inhibit motility and invasion of two astrocytoma cell lines (SF268 and U87) [177]. These results demonstrate a promising efficacy of the drug for the treatment of GBM. However, more studies are needed.

One mechanism that allows GBM to be chemoresistant is the active efflux of these anticancer drugs across the cell membrane. Overexpression of ATP-binding cassette transporters (ABC Transporters) plays a crucial role in generating the multidrug resistant phenotype in GBM. P-glycoprotein or ABCB1 protein (ATP Binding Cassette Subfamily B Member 1) is an important member of the ABC transporter family, primarily responsible for GBM resistance to temozolomide. Inhibition of ABCB1 allows for greater efficacy of temozolomide, leading to an increase in GBM cell death [178]. Thus, the combination of ABC transporter inhibitors with other drugs may be a strong strategy to increase the survival of patients diagnosed with GBM. 

The combined treatment of Mifepristone and temozolomide was able to reduce the expression of P-glycoprotein and increase the concentration of temozolomide in the brains of rats. The combined treatment was able to double the survival of 70% of the rats [169]. Endoxifen, the most clinically relevant metabolite of tamoxifen, is a substrate of P-glycoprotein. Oral and systemic administration of endoxifen in Abcb1a/1b(−/−) mice showed increased brain concentration when compared to wild-type mice [179].

The efflux transporter ATP-binding cassette subfamily G member 2 (ABCG2), also referred to as breast cancer resistance protein, is a member of the ATP-binding cassette subfamily G. Lin et al. demonstrate that the expression of ABCG2 may regulate the TMZ resistance, and its inhibition could improve brain infiltration by the drug as showed by Gooijer et al. [180,181]. Ko143 is an inhibitor of ABCG2 and could be a future option for GBM treatment (Table 1). The Lustig et al. research suggests that the drug could improve TMZ efficacy and may be a great inhibitor for P-glycoprotein. Additionally, as Gaber et al. discovered, ABCG2 reductions in photosensitizer accumulation may aid GBM stem cells avoid phototoxic death. During photosensitizer accumulation and irradiation, the block of ABCG2 by ko143 appears to restore entire photodynamic treatment sensitivity of this cell type [182,183]. Despite this evidence, further studies will be required to obtain new conclusions about the drug.

**Table 1 cancers-14-03203-t001:** Molecular structure, mechanisms of action and efficacy of new drugs tested in GBM in vitro or in vivo.

Name	Structure	Molecular Formula	Mechanism	Overall Survival (OS)	References
Tamoxifen	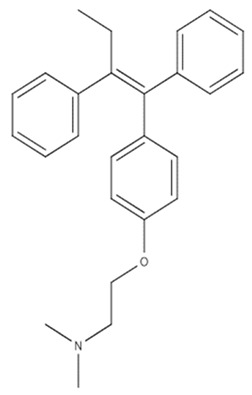	C_29_H_35_NO_2_	OHT causes GBM cell death via an autophagy-related mechanism.	17.5 months	[178,179]
Agathisflavone	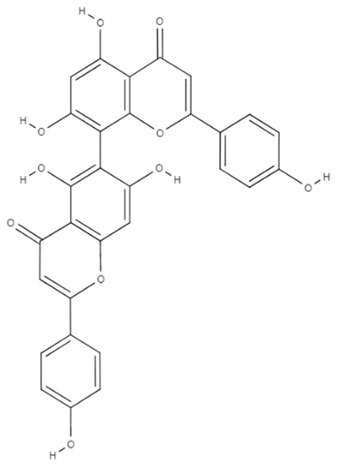	C_30_H_18_O_10_	The drug decreased microglia neuroinflammation and reduced microglia and neuron neurotoxicity.	Not tested in humans	[176]
Mifepristone	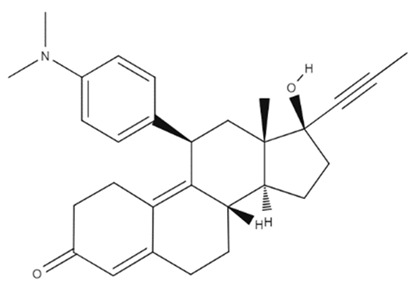	C_29_H_29_NO	The drug binds to progesterone and to glucocorticoids receptors. Resulting in a reduction of progesterone-dependent genes.	Not tested in humans	[169]
Metformin	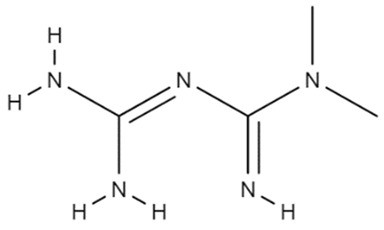	C_4_H_11_N_5_	The drug down-regulates SOX2 expression in vitro, reduces the formation ability of glioblastoma cells, and blocks GBM xenograft growth in vivo.	Not prolong OS	[176,177]
Ko143	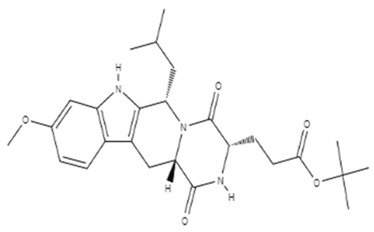	C_26_H_35_N_3_O_5_	The drug could improve TMZ efficacy and may be a great inhibitor for P-glycoprotein.	Not tested in humans	[182,183]

## 7. Conclusions

GBM is the most common and aggressive brain tumor, with a 15-month survival rate for patients. This tumor is highly invasive, with large necrosis and hypoxia areas, modifying the surrounding neuronal tissue. GBM is very heterogeneous, with several infiltrated cell types, and many subcellular populations of tumor cells. The main factor for this intrinsic heterogeneity is subpopulations of tumor stem cells capable of tumor initiation, asymmetric division and chemoresistance. Despite all the studies about GBM, the prognostic remains extremely bad, and there are just a few therapeutic approaches approved for the treatment of GBM. Modulation of BBB permeabilization, the use of oncolytic virus and drugs or other approaches that modulate the tumor cytoskeleton and tumor stem cells, seem to be good therapeutic targets for the treatment of this tumor. 

## Figures and Tables

**Figure 1 cancers-14-03203-f001:**
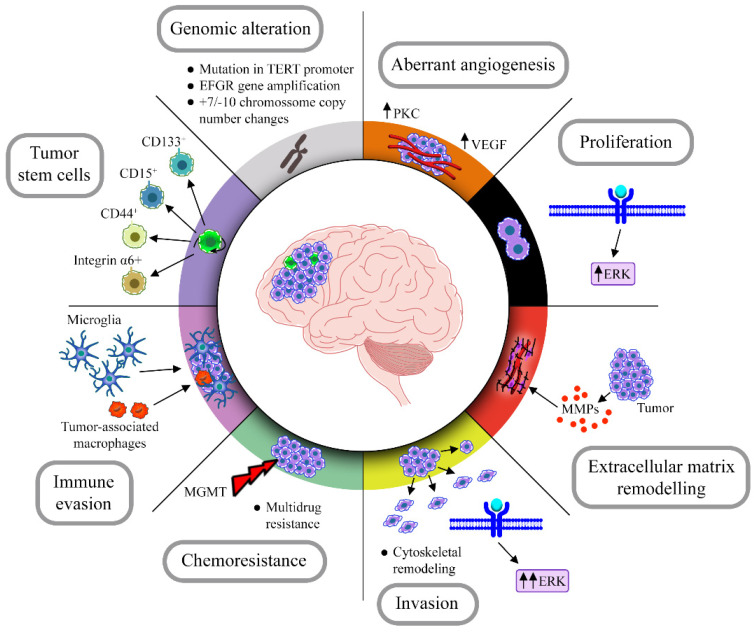
Hallmarks of GBM. GBM has some striking features that contribute to its aggressive phenotype and that could be therapeutic targets. Infiltrating cells, genomic alterations, aberrant angiogenesis, immune evasion and modulation of the extratumoral environment are some of the characteristics present in the vast majority of these tumors.

**Figure 2 cancers-14-03203-f002:**
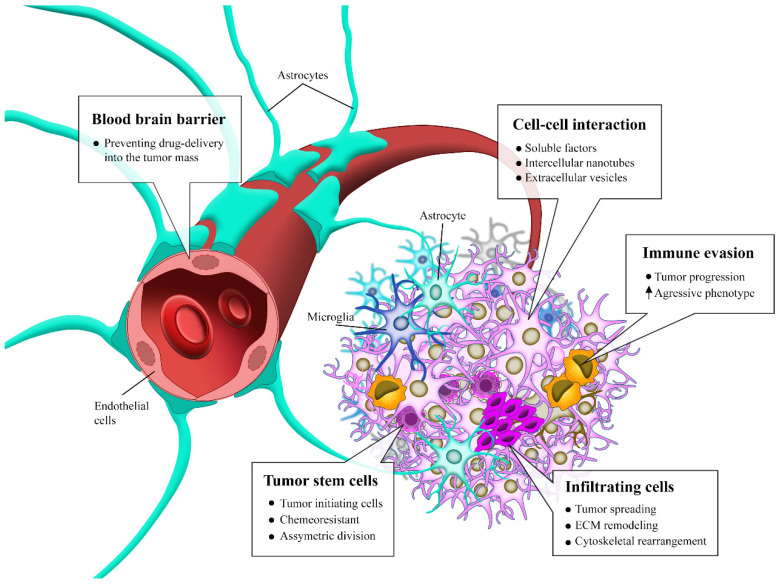
Challenges in GBM treatment. GBM has remarkable features that make it so malignant and difficult to treat. Infiltrative cells, immune evasion, stem cells capable of repopulating the tumor mass after treatment and exchange of soluble factors and vesicles among the tumor cells, form a complex microenvironment that, together with the blood–brain barrier that hinders the entry of molecules into the brain tissue reducing the pharmacological options for the treatment of this tumor, contributes to the difficulty in the therapeutic advance of GBM.

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
