# Peer review of "Obstacles to Glioblastoma Treatment Two Decades after Temozolomide"

_cancers, 2022, doi:10.3390/cancers14133203_

Round 1
Reviewer 1 Report
This is a well written narrative review on current difficulties and therapeutic research in GBM. This work probably adds very little data to a scholar who is expert scholar on the topic, not all the evidences on GBM research are mentioned, with the review missing a systematic approach, which is obviously impossible in such a wide topic. Nonetheless, the review is well organized and easy to read and in my opinion it gives all the basic data needed to approach the topic to less expert readers.
Author Response
Thank you very much for reviewing the manuscript. We agree that the topic is very broad and this makes it difficult to include all the evidence. However, we believe that it can be an excellent material for beginners in the subject.
Reviewer 2 Report
“Obstacles to Glioblastoma Treatment Two Decades after Temozolomide” by Cruz et al. provides very important facts regarding glioblastoma treatment obstacles. It is a comprehensive, well-written review paper that could be high-interesting to Cancers MDPI audience. Although I have words of praise for this work, it would benefit if some important facts are included:
Major concerns:
1. The role of ABC transporters is completely neglected in this work. However, ABC transporters should be reviewed in the context of glioblastoma stem cells and the blood-brain barrier. Besides, temozolomide and many anticancer drugs reviewed in this paper are substrates for ABC transporters. There are many ABC transporters’ inhibitors used to modulate glioblastoma multidrug resistance and overcome the obstacle of the blood-brain barrier that deserve to be discussed.
2. It is not clear why the authors gave special attention to mifepristone, agathisflavone, and tamoxifen.
3. One can see that some of the authors recently published a research article on agathisflavone and this is the only work that considers agathisflavone as an anti-glioma agent (Nascimento RP, Dos Santos BL, da Silva KC, Amaral da Silva VD, de Fátima Costa M, David JM, David JP, Moura-Neto V, Oliveira MDN, Ulrich H, de Faria Lopes GP, Costa SL. Reverted effect of mesenchymal stem cells in glioblastoma treated with agathisflavone and its selective antitumoral effect on cell viability, migration, and differentiation via STAT3. J Cell Physiol. 2021 Jul;236(7):5022-5035. doi: 10.1002/jcp.30209. Epub 2020 Dec 23. PMID: 33368262.).
4. Mifepristone interacts with P-glycoprotein (an ABC transporter) and this is also not mentioned (Llaguno-Munive M, León-Zetina S, Vazquez-Lopez I, Ramos-Godinez MDP, Medina LA, Garcia-Lopez P. Mifepristone as a Potential Therapy to Reduce Angiogenesis and P-Glycoprotein Associated With Glioblastoma Resistance to Temozolomide. Front Oncol. 2020 Oct 5;10:581814. doi: 10.3389/fonc.2020.581814. Erratum in: Front Oncol. 2021 Mar 22;11:675806. PMID: 33123485; PMCID: PMC7571516.)
5. Tamoxifen also interacts with P-glycoprotein and a combination of tamoxifen and temozolomide should be considered as a combination of two P-glycoprotein substrates where one of the substrates takes on the role of inhibitor and increases the intracellular concentration of the other substrate applied in combination.
Minor concerns:
1. Convection-enhanced delivery (CED) is explained twice (in 4. Strategies to overcome the blood-brain barrier in GBM treatment and in 5. Using oncolytic viruses to reach tumor cells)
2. The authors stated “Importantly, the cell therapies, as well as the treatment with chimeric antigen receptor (CAR) T cell represent a new treatment modality with exciting potential, although it has not yet been clinically effective [165]. Therefore, its relevance to the treatment of brain tumors is evident.” The second sentence is not true and needs to be rephrased.
3. DELYTACT (teserpaturev/G47∆ a genetically engineered oncolytic HSV-1) has received conditional and time-limited approval in Japan for the treatment of patients with malignant gliomas based on the phase 2 study results. I think this is worth mentioning in this paper.
Author Response
Dear Reviewers,
Thank you very much for your comments and suggestions about this manuscript. We agree that your concerns are appropriate and have substantially improved the way we present our review. Please find our answers to your questions and comments below and the new version of the manuscript we are presenting attached to the Cancers website.
Reviewer 2:
Comments and Suggestions for Authors
“Obstacles to Glioblastoma Treatment Two Decades after Temozolomide” by Cruz et al. provides very important facts regarding glioblastoma treatment obstacles. It is a comprehensive, well-written review paper that could be high-interesting to Cancers MDPI audience. Although I have words of praise for this work, it would benefit if some important facts are included:
Major concerns:
- The role of ABC transporters is completely neglected in this work. However, ABC transporters should be reviewed in the context of glioblastoma stem cells and the blood-brain barrier. Besides, temozolomide and many anticancer drugs reviewed in this paper are substrates for ABC transporters. There are many ABC transporters’ inhibitors used to modulate glioblastoma multidrug resistance and overcome the obstacle of the blood-brain barrier that deserve to be discussed.
We agree with the reviewer. Due to the broad scope of the topic of our review, we ended up not writing about ABC transporters. To fill this gap, we have added paragraphs on the effect of some drugs on ABC transporters in the new version of the manuscript. Now, on page 18 we added: “One mechanism that allows glioblastoma to be chemoresistant is the active efflux of these anticancer drugs across the cell membrane. Overexpression of ATP-binding cassette transporters (ABC Transporters) plays a crucial role in generating the multidrug resistant phenotype in glioblastoma. P-glycoprotein or ABCB1 protein (ATP Binding Cassette Sub-family B Member 1) is an important member of the ABC transporter family, primarily re-sponsible for glioblastoma resistance to temozolomide. Inhibition of ABCB1 allows for greater efficacy of temozolomide, leading to an increase in glioblastoma cell death [184]. Thus, the combination of ABC transporter inhibitors with other drugs may be a strong strategy to increase the survival of patients diagnosed with glioblastoma.”
- It is not clear why the authors gave special attention to mifepristone, agathisflavone, and tamoxifen.
We agree with the reviewers. We unintentionally draw attention to these three drugs in particular. In the new version of the manuscript, we cut some of the text about these drugs and included all the drugs in a single topic.
- One can see that some of the authors recently published a research article on agathisflavone and this is the only work that considers agathisflavone as an anti-glioma agent (Nascimento RP, Dos Santos BL, da Silva KC, Amaral da Silva VD, de Fátima Costa M, David JM, David JP, Moura-Neto V, Oliveira MDN, Ulrich H, de Faria Lopes GP, Costa SL. Reverted effect of mesenchymal stem cells in glioblastoma treated with agathisflavone and its selective antitumoral effect on cell viability, migration, and differentiation via STAT3. J Cell Physiol. 2021 Jul;236(7):5022-5035. doi: 10.1002/jcp.30209. Epub 2020 Dec 23. PMID: 33368262.).
The reviewer is correct. Despite being an interesting drug and looking promising, this is the only article that considers agathisflavone an anti-glioma drug. In order to draw less attention to agathisflavone, we removed some of the text and added a sentence saying that it has not been tested in humans and more studies are needed to prove its effectiveness as an anti-glioma agent. Now on page 17 we wrote: “Despite looking very promising for use as an anti-glioma drug, agathisflavone has not yet been tested in humans and further studies are still needed to determine its true effective-ness.”
- Mifepristone interacts with P-glycoprotein (an ABC transporter) and this is also not mentioned (Llaguno-Munive M, León-Zetina S, Vazquez-Lopez I, Ramos-Godinez MDP, Medina LA, Garcia-Lopez P. Mifepristone as a Potential Therapy to Reduce Angiogenesis and P-Glycoprotein Associated With Glioblastoma Resistance to Temozolomide. Front Oncol. 2020 Oct 5;10:581814. doi: 10.3389/fonc.2020.581814. Erratum in: Front Oncol. 2021 Mar 22;11:675806. PMID: 33123485; PMCID: PMC7571516.)
In the new version of the manuscript, we added this information in section 6. On page 18 we wrote: “The combined treatment of Mifepristone and temozolomide was able to reduce the expression of P-glycoprotein and increase the concentration of temozolomide in the brains of rats. The combined treatment was able to double the survival of 70% of the rats [174]. Endoxifen, the most clinically relevant metabolite of tamoxifen, is a substrate of P-glycoprotein. Oral and systemic administration of endoxifen in Abcb1a/1b(−/−) mice showed increased brain concentration when compared to wild-type mice”
- Tamoxifen also interacts with P-glycoprotein and a combination of tamoxifen and temozolomide should be considered as a combination of two P-glycoprotein substrates where one of the substrates takes on the role of inhibitor and increases the intracellular concentration of the other substrate applied in combination.
In the new version of the manuscript, we added this information in section 6. On page 18 we wrote: “The combined treatment of Mifepristone and temozolomide was able to reduce the expression of P-glycoprotein and increase the concentration of temozolomide in the brains of rats. The combined treatment was able to double the survival of 70% of the rats [174]. Endoxifen, the most clinically relevant metabolite of tamoxifen, is a substrate of P-glycoprotein. Oral and systemic administration of endoxifen in Abcb1a/1b(−/−) mice showed increased brain concentration when compared to wild-type mice”
Minor concerns:
- Convection-enhanced delivery (CED) is explained twice (in 4. Strategies to overcome the blood-brain barrier in GBM treatment and in 5. Using oncolytic viruses to reach tumor cells).
We corrected this error in the text of the new version of the manuscript.
- The authors stated “Importantly, the cell therapies, as well as the treatment with chimeric antigen receptor (CAR) T cell represent a new treatment modality with exciting potential, although it has not yet been clinically effective [165]. Therefore, its relevance to the treatment of brain tumors is evident.” The second sentence is not true and needs to be rephrased.
We corrected this error in the text of the new version of the manuscript.
- DELYTACT (teserpaturev/G47∆ a genetically engineered oncolytic HSV-1) has received conditional and time-limited approval in Japan for the treatment of patients with malignant gliomas based on the phase 2 study results. I think this is worth mentioning in this paper.
We added this information in the text of the new version of the manuscript (topic 5: Using oncolytic viruses to reach tumor cells page 14).
Reviewer 3 Report
The authors give a general overview of glioblastomas and their treatment in this manuscript. In most sections, the authors only illustrate the critical facts and findings. However, no further information is given. In addition, the information in quite many sections is fragmented. The authors should reorganise the information. Otherwise, it will affect the reading experience.
1. What is GBM? I suppose it is glioblastomas. The short form suddenly appears in the first paragraph without explanation.
2. On page 3, the authors mention that chromosome aberration is an important feature. The authors should give more details, such as what kind of the aberration and how the aberration will affect the aggressiveness of glioblastomas?
3. On page 4, the authors mention that various drugs such as TMZ, paclitaxel, carboplatin, and ectoposide have been used for the treatment. The authors may make a table to illustrate the efficacy of various drugs. So the readers will have a general view of the effectiveness of the current treatment.
4. On page 5, the authors mention IDH mutation is a good prognostic factor. What are the reasons? The information on pages 5 – 7 is a bit fragmented. The authors should consider reorganising the information.
5. On page 10, the author should illustrate the method used for transiently opening the blood brain barrier. Similar to point 4, the information in this section should be reorganised. The flow of the manuscript is not smooth
6. On page 12, the authors mention that direct IT inoculation is the most successful in the FDA-approved T-Vec therapy. The authors should give numerical evidence to support this statement. In addition, the author should briefly illustrate why repeated IT inoculation is not feasible.
7. On page 14, the authors mention there is a table, table 1. However, I cannot find it.
Author Response
Answers to Reviewer:
Dear Reviewers,
Thank you very much for your comments and suggestions about this manuscript. We agree that your concerns are appropriate and have substantially improved the way we present our review. Please find our answers to your questions and comments below and the new version of the manuscript we are presenting attached to the Cancers website.
Reviewer 3:
Comments and Suggestions for Authors
The authors give a general overview of glioblastomas and their treatment in this manuscript. In most sections, the authors only illustrate the critical facts and findings. However, no further information is given. In addition, the information in quite many sections is fragmented. The authors should reorganise the information. Otherwise, it will affect the reading experience.
- What is GBM? I suppose it is glioblastomas. The short form suddenly appears in the first paragraph without explanation.
GBM is the abbreviation of glioblastoma. In the new version of the manuscript, we corrected this information
- On page 3, the authors mention that chromosome aberration is an important feature. The authors should give more details, such as what kind of the aberration and how the aberration will affect the aggressiveness of glioblastomas?
In the new version of the manuscript, we added on page 2: “The most common cromossomal abnormalities found in GBMs are gain of chromosome 7p (trisomy) and loss of chromosome 10q (monosomy). One of the oncogenes on 7p is EGFR, that is amplified in about one third of GBMs, and in 10q there is the inactivation of tumor-supressor genes in the telomeric region of 10q, specially PTEN”.
- On page 4, the authors mention that various drugs such as TMZ, paclitaxel, carboplatin, and ectoposide have been used for the treatment. The authors may make a table to illustrate the efficacy of various drugs. So the readers will have a general view of the effectiveness of the current treatment.
In the new version of the manuscript, we have added a table with the molecular structures of drugs and their efficacy.
- On page 5, the authors mention IDH mutation is a good prognostic factor. What are the reasons? The information on pages 5 – 7 is a bit fragmented. The authors should consider reorganising the information.
We agree with the reviewer and reformulated this part of the text in the new version of the manuscript.
- On page 10, the author should illustrate the method used for transiently opening the blood brain barrier. Similar to point 4, the information in this section should be reorganised. The flow of the manuscript is not smooth
We agree with the reviewer and reformulated this part of the text in the new version of the manuscript.
- On page 12, the authors mention that direct IT inoculation is the most successful in the FDA-approved T-Vec therapy. The authors should give numerical evidence to support this statement. In addition, the author should briefly illustrate why repeated IT inoculation is not feasible.
We agree with the reviewer and we added more information this part of the text in the new version of the manuscript.
- On page 14, the authors mention there is a table, table 1. However, I cannot find it.
In the new version of the manuscript, we have added a table with the molecular structures of drugs and their efficiency.
Round 2
Reviewer 2 Report
The authors accepted all suggestions and improved the quality of their work. Therefore, I recommend the manuscript for publication in its present form.
Reviewer 3 Report
The authors have already addressed all of the concerns and improved the manuscript.